

# Skewing angle magnet and coil reduced starting torque in a permanent magnet synchronous generator for a small vertical axis wind turbine

Priwan Pongwan[1], Kusumal Chalermnayanont[2], Mintra Trongtorkarn[3], Suppachai Jina[1],

Montri Luengchavanon[4*]

[1] Energy Technology Program, Faculty of Engineering, Prince of Songkla University, Hatyai, Songkhla 90110 Thailand.

[2] Department of Electrical Engineering, Faculty of Engineering, Prince of Songkla University, Hatyai, Songkhla 90110 Thailand.

[3] Department of Industrial Electrical Technology, Faculty of Science and Technology, Suratthani Rajabhat University, Suratthani  84100, Thailand.

[4*] Sustainable Energy Management Program, Faculty of Environmental Management, Prince of Songkla University, Hatyai, Songkhla 90110 Thailand.

*Correspondence to*: Montri Luengchavanon ([montri.su@psu.ac.th](mailto:montri.su@psu.ac.th))


**Abstract.** This work investigated the effects of changing the skewing angle of a magnet coil on starting torque in a permanent magnet generator (PMSG) fitted in a low speed vertical wind turbine. The optimal skew angle of the magnet-coil was found to be 15-0 (degrees), generating 1.22 (N-m) starting torque and 295.40 (W) compared with a skew angle of 0-0 (degrees). This skew angle reduced starting torque and power by 5.43% and 1.96%, respectively. A Savonius and H-Darrieus stacked turbine

blade operated at a wind speed of 1.90 m/s and 1.31 N-m torque. This blade was used in a fully operational vertical wind turbine, was connected to the PMSG that can cut-in speed of 2.1 m/s. It was concluded that a 15-0 (degree) skewing angle magnet-coil can be applied to a low speed vertical wind turbine.

## Introduction

25       Wind energy is a relatively inexpensive renewable energy resource, which can be widely distributed. In particular, due to the available space and wind currents, ocean environmental conditions provide a suitable location to harness wind energy. In Europe, offshore wind energy is continuously being developed, with net offshore wind energy power capacity (2649 MW) and cumulative offshore wind capacity (approx. 18,499MW) reported to have increased in 2018 (Walsh, 2019).





Bloomberg New Energy Finance have suggested that the global offshore wind market will approach a cumulative capacity of
115GW, with a 16% annual growth rate from 2017 to 2030 (Jiang et al., 2020). Therefore, the development of cost effective
megawatt wind turbines are likely to become more popular across Europe and Asia, with costs likely to be reduced via two
methods. In utilizing future offshore wind energy, the vertical axis wind turbine (VAWT) is expected to perform better than
the horizontal axis wind turbine (HAWT). This is because the VAWT has a lower center of gravity and simple mechanical
structure, which is easy to fabricate, and requires low maintenance. Moreover, the VAWT has a high potential to be scaled to
a number of different sizes (Paquette and Barone, 2012).

In order to meet commercialization requirements, VAWT technology is constantly improving. Due to its unique
rotating motion, the hardness of the VAWT blade provides the greatest amount of wind power on the upwind half
circumference, however it provides a lower power coefficient compared with HAWT (Liu et al., 2019). This has led to several
investigators exploring ways to improve the power output of isolated wind turbines by optimising and developing variable
pitch technology or air-foils. The power coefficient can be improved by harnessing more wind energy via auxiliary devices,
such as deflectors and diffusers (Takao et al., 2009;Wong et al., 2018;Malipeddi and Chatterjee, 2012). The design of new
guide vane geometry for a VAWT has been shown to improve power with high tip speed ratios (Takao et al., 2009). The
diffuser structure around a VAWT allows for an increased power coefficient and more energy from wind turbines positioned
adjacent to each other (a wind turbine array) providing a new approach to improve the power output of VAWT's (Dabiri,
45    2011).

The electrical generator is the main device of wind turbine technology, and assists with providing higher efficiency.
Presently, more than 90% of wind power plants have an electrical power range between 0.1 – 20 kW, and are often used in
permanent magnet synchronous generators (PMSG) (Soderlund et al., 1996). As these wind power plant generators are driven
directly from the shaft of wind turbine using no reduction gear (Haring et al., 2003), low speed generators can sufficiently
provide a large number of poles on the rotor, with a relatively small pole pitch (Cistelecan et al., 2007). However, there are a
number of technological problems related to stator generator production, and the automatically generated process of the three-
phase (m=3) fabricated winding insertion. Therefore, the application of generators, which use windings with a number of poles
on poles and phase q=1, can be limited. The relationship between pole pair number $p$ and gear teeth number $Z_1$ is defined in
Equation (1) (Germar et al., 2012).

55          $Z_1=2pmq$                              (1)

Where $m$ is number of phases, and $q$ is number of teeth on the pole and phase. When the phase is q=1, the system needs 6 teeth
divisions to enable the winding to settle down on one rotor's polar branch. The increased attention to the winding possessing
sufficient length, with overlapping frontal connections, has led to an increase in growth, with an increase in copper
consumption and eddy current losses. This is a disadvantage, has it has led to a synchronous generator with non-overlapping
concentration windings for wind power (Saavedra et al., 2014;Levin et al., 2012;Magnussen and Sadarangani, 2003). Based
on the PMSG generator producing high starting torque values, and a dependency of the torque to be related to the rotor's
rotation angle (Bakiev et al., 2018). The skewing ways of a permanent magnet machine can improve starting torque, by



affecting the torque and air-gap flux shape. However, the starting torque can also be reduced when the electrical power is decreased (Lateb et al., 2006), leading to a compromise with the skewing technique when it is applied to small wind turbines.

65  This study investigated the angle of skewing magnet-coils for reducing starting torque and electrical power in a PMSG. The optimal angle of the skewing magnet-coil was applied to a vertical wind turbine to validate the operation of a low speed vertical wind turbine at 2 m/s.

**Experiments**

70  In order to conduct the experiments of this investigation, a suitable test environment was arranged, including obtaining a laboratory machine. The acrylic plate was automatically changed to adjust the skewing angle to support the magnets and coils at 0-0, 0-5, 0-10, 0-15, 5-0, 5-5, 5-10, 5-15, 10-0, 10-5, 10-10, 10-15, 15-0, 15-5, 15-10, 15-15 degrees, respectively; as shown in **Figure 1**. The testing station was conducted in two parts, firstly, the controller and PMSG were used for the 3 phase inverter (1.5 kW/220 Vac, Input: 220 Vac, Output: 380 Vac, Jaden) in order to vary the speeds of 3 hp motors (0.78 kW, VEB

75 elektromotorenwerk). The PMSG was then conducted with 12 magnets (NdFeB), which were used to skew the angles (sizes was 100x20x5 mm: 251 mT) that were installed onto the acrylic disc (size: 400x10 mm) for the rotor section. The 9 coils (size was 100x70x12 cm) and air core (2x3.5x2 cm) were installed onto the acrylic disc (size: 400x10 mm) for the stator section, and the PMSG was used with a 2 mm air-gap (**Figure 2** and **Table 1**). Secondly, the measurement instruments included a torque meter (BCM sensor technologies, Model: 1811, Cap: 500 Nm, Accuracy (torque): 0.5% fs, Supply: ±15 Vdc, Max:

80 speed 7000 rpm, Load current < 10mA), and a power measurement, using a 3 phase Power & Harmonic Analyzer (Chauvin Arnoux: CA8331). The PMSG output was connected to a 3 phase switch and variable 300 W resistance load.

**Table 1.** The PMSG parameters

| Constructive elements | value | Unit |
|---|---|---|
| Number of phase | 3 | - |
| Number of pole (P) | 12 | - |
| Number of stator slots | 9 | - |
| Flux density of NbFeB Magnet | 251 | mT |
| Magnet length | 100x20x12 | mm |
| Coil length | 100x70x12 | mm |
| Number of coils | 2700 | N/Phase |
| Rotor outer diameter | 400 | mm |
| Stator outer diameter | 400 | mm |
| Air gap | 3 | mm |



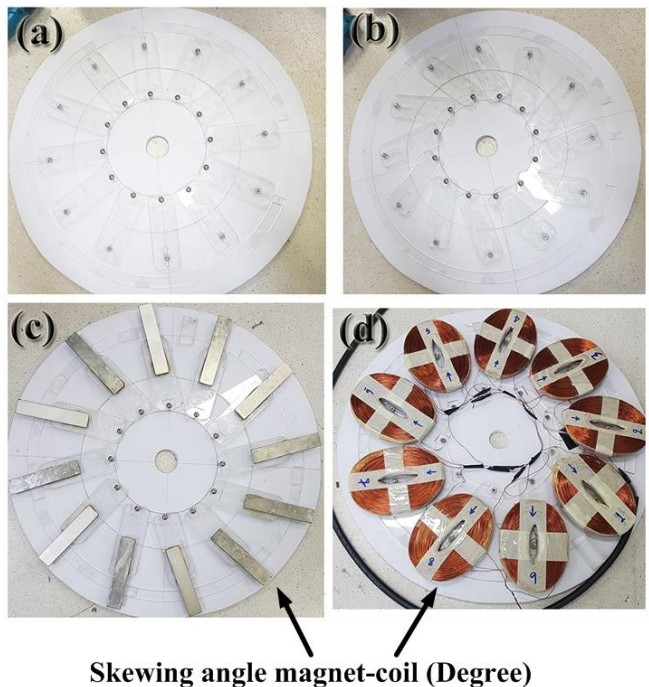

**Figure 1:** (a) the acrylic plate for automatically skewing the angle of the magnets; (b) the acrylic plate can automatically skew the angle of the coils; (c) the 12 magnets installed on the acrylic disc for the rotor section; (d) 9 coils installed on the acrylic disc for the stator section.

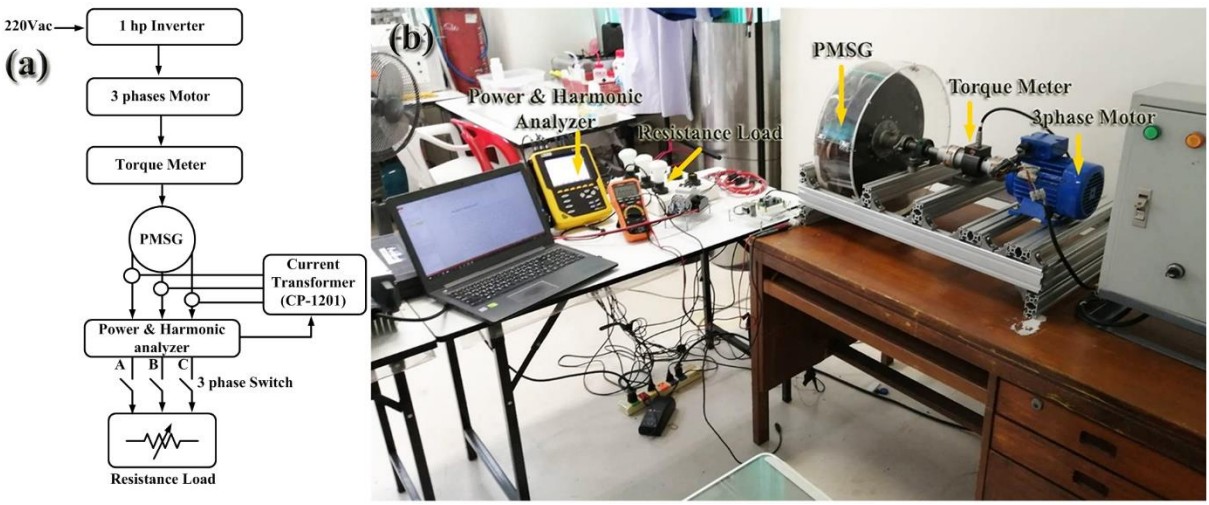

**Figure 2**: (a) Diagram showing the measurement of the Permanent Magnet Synchronous Generator (PMSG) connected to the motor driver and electrical load; (b) the PMSG experimental testing station.




**Results and discussion**

**Fig 3 (a)** shows the comparison of the output torque with the skewing angle magnets-coils under no-load conditions.
The trends fluctuated, with torque ranging between 0.05-0.30 N-m. The skewing angle of the magnets-coils could not exactly

indicate a reduced torque. **Fig 3 (b)** shows the comparison of the output torque with the skewing angle magnets-coils under on

load conditions. The trends indicated that the transient torque was related to the time duration and the skewing angle of the

magnets-coils. The starting torque was increased from 0.20 to 1.4 N-m over 2.5 seconds, however after 4 seconds, the generated

torque remained constant, with the behavior of each trend dependent on the skewing angle of the magnets-coils. The PMSG

was connected to the electrical load that fabricated the induction of power between the magnets and coils. It was found that

the trends between on load and no load were different, with the PMSG generating power induction between the magnets and

coils when operated with no load. The electricity transferred from coils (stator) without load has previously been shown to

fluctuate the signal (Bülow et al., 2012). In addition, the starting torque partly depends on the load, which breaks the torque

produced on the rotor's rotation angle (Ose-Zala et al., 2014). These starting torques could be fabricated over 2.5 seconds.


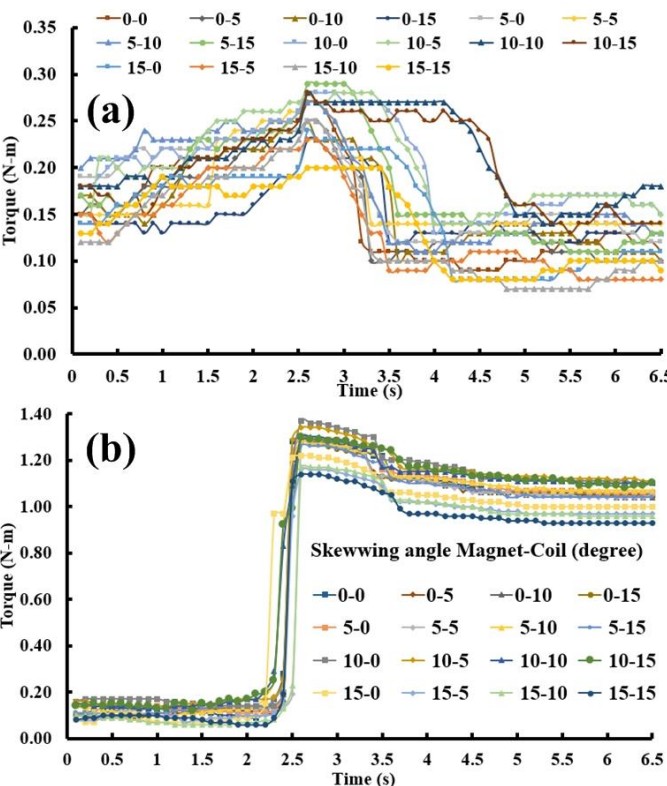

**Figure 3:** The behavior of PMSG based on starting torque (a) the transient starting torque of PMSG no-load conditions, (b) the transient
starting torque of PMSG with 300W electrical load.



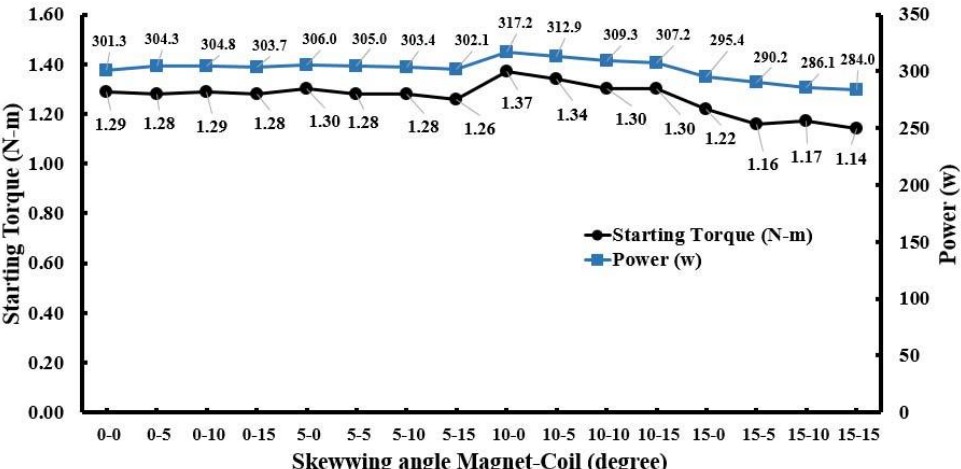

**Figure 4:** The relationship between starting torque (N-m) and skewing angle magnet-coil (degree) which compared maximum starting torque and electrical power.

**Fig 4** shows comparison values between the starting torque and power when changing the skewing angle magnet-coils in the PMSG generator. In the control condition, it was determined that the skewing angle magnet-coil at 0-0 generated 1.29 N-m and 301.3 W compared with other conditions. The skewing angle magnet-coil at 0-5, 0-10, 0-15, 5-0, 5-5, 5-10, 5-15 degrees provided similar starting torque and power values to 0-0 degrees. In contrast, the skewing angle of the magnet-coil at 10-0, 10-5, 10-10, 10-15, 15-0, 15-5, 15-10 and 15-15 degrees indicated that 10-0 degree produced a higher starting torque (1.37 N-m) and power (317.2 W), respectively, with both parameters gently decreasing until 15-15 degree. In particular, starting torque at a skewing angle of 15-0 (1.22 N-m), 15-5 (1.16 N-m), 15-10 (1.17 N-m) and 15-15 degrees (1.14 N-m), were lower than the 0-0 degree condition. These conditions are applicable to a wind turbine.

The starting torque ($T_{starting}$) can be fabricated by estimating the energy in the air-gap by stepping the rotor. The energy changed in the PMSG generator and iron is minimal when compared to the air (Eom et al., 2001).

$$T_{starting}(\theta) \approx -\frac{\partial W_{air-gap}(\theta)}{\partial \theta} = -\frac{1}{2}\emptyset_g^2\frac{dR}{d\theta} \qquad (1)$$

Where $W_{air-gap}(\theta)$ is energy within the air-gap, $\phi_g$ is the air flux between magnet and coil, $\theta$ is rotor angular position in electrical degrees and $R$ is the air-gap reluctance. **Eq (1)** can be expressed in component form for a radial flux permanent magnets (RFPM) generator (Eom et al., 2001), the $\phi_g$ is the main factor that affected to the skewing angle magnet-coil. **Eq (2)** can be expressed in the air flux from the magnet to coil (Wirtayasa et al., 2017).

$$\emptyset_g = \frac{2}{\pi}B_{mg}\frac{\pi}{2p}(R_{out}^2 - R_{in}^2) \qquad (2)$$

Where $B_{mg}$ is flux density from magnet, $2P$ is the number of poles, $R_{in}$ and $R_{out}$ are the radial radius of the permanent magnet at the inner and outer diameter. Therefore, the skewing angle magnet-coil can be affected from the area of $\phi_g$, $R_{in}$ and $R_{out}$ by increasing or decreasing the starting torque ($T_{starting}$) in the PMSG; as shown in **Fig 5**. Additionally, the skewing angle magnet-



coil is also affected by the electrical power ($P_{out}$) of the PMSG, causing a change in the voltage (rms) value of the $E_{EMF}$ in the PMSG; as shown in **Eq (3-4-5)**.

$$k_{w1} = k_{d1}k_{p1}k_{sn1} \tag{3}$$

$$E_{EMF} = \pi\sqrt{2}fN_1k_{w1}\emptyset_g \tag{4}$$

$$P_{out} = \frac{E_{EMF}^2}{(R_m+R_L)^2+X_S^2}R_L \tag{5}$$

Where $k_{w1}$ is the winding factor of the coil, $k_{d1}$ is the distribution factor, $k_{sn1}$ is the skew factor, $f$ is frequency, $N_1$ is
the number of armature turns in one phase, $R_m$ is the armature resistance, $R_L$ is load resistance, and $X_S$ is the synchronous reactance. Based on **Eq (3)** is represented that the $k_{w1}$ is affected from the shape and winding of the coil. While $k_{sn1}$ is the main factor, which changed starting torque and power when skewing the angle magnet-coil.

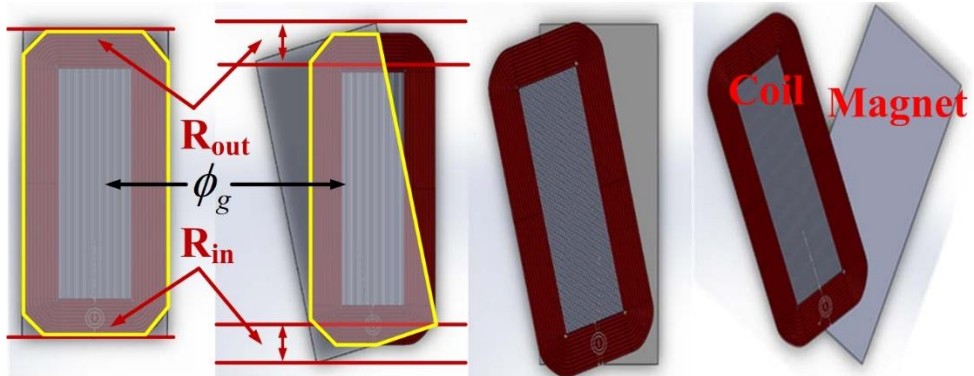

**Figure 5:** The skewing angle magnet-coil affected the air flux between the magnet and coil and radial radius of the permanent magnet at the ($R_{in}$) inner and ($R_{out}$) outer diameter.

(Suppachai et al., 2019 ) reported that the Savonius and H-Darrieus blade, which are stacked together, can operate at 10kW wind turbine; as shown in **Figure 6**. **Table 1** shows the characteristics under these conditions for selecting the optimal
skewing angle of the magnet-coil condition, and the properties of the blade are shown in **Figure 7**. In order to initiate the first turn of the wind turbine, it was determined that the torque of the blade must be higher than the starting torque from the PMSG generator, with the wind turbine requiring to cut-in at approximately 2 m/s condition. The blade can be focused to a wind speed of 1.90 m/s and 1.31 N-m to be matched with the starting torque of the PMSG generator. The skewing angle magnet is easier to fabricate compared with skewing angle coil for the PMSG generator (Bianchi and Bolognani, 2002). However, since
the PMSG generator requires a high power with a reduced starting torque, an optimal skewing angle magnet-coil of 15-0 degrees enables the production of 1.22 N-m and 295.40 W, to reduce starting torque (5.43%) and power (1.96%), respectively.



The connection of the blade and PMSG generator, which allows the wind turbine to cut-in at low speed wind, $J_{cut\text{-}in}$ (equivalent inertia) is shown in **Eq (6)** (Belmili et al., 2017;Hsieh et al., 2009).

$$J_{Cut\text{-}in} = T_{blade} - T_{starting} \qquad (6)$$


Where $T_{blade}$ is torque from the blade of the turbine (**Table 2**). The cut-in of this vertical wind turbine cannot start turning at 1.90 m/s wind speed due to the systems combined force of $T_{starting}$. Hence, the real operation of this vertical wind turbine system can be cut-in at 2.1 m/s.

**Table 2:** The comparison of parameters between the PMSG generator and Savonius and H-Darrieus blade, which were stacked together for selected the skewing angle magnet-coil condition.

| PMSG generator As shown in **Figure 2** | Parameters | | | |
|---|---|---|---|---|
| Starting Torque (N-m) | Skewing angle magnet-coil (Degree) | Electrical Power (W) | % Reduced starting torque | % Reduced power |
| 1.29 | 0-0 | 301.30 | - | - |
| **1.22** | **15-0** | **295.40** | **5.43** | **1.96** |
| 1.16 | 15-5 | 290.20 | 10.08 | 3.68 |
| 1.17 | 15-10 | 286.10 | 9.30 | 5.04 |
| 1.14 | 15-15 | 284.00 | 11.63 | 5.74 |
| Blade of Savonius and H-Darrieus stacked together as shown in **Figure 6 (a)** and **Figure 7** | Parameters | | | |
| Torque (N-m) | Wind Speed (m/s) | - | - | - |
| 0.22 | 1.52 | - | - | - |
| 0.46 | 1.60 | - | - | - |
| 0.98 | 1.75 | - | - | - |
| **1.31** | **1.90** | **-** | **-** | **-** |

**Figure 8** shows the comparison of the 0-0 and 15-0 degrees skewing angle magnet-coil conditions, based on the relationship of the power and rotation speed. The skewing angle magnet-coil at 15-0 degrees was selected for using in the

PMSG generator. The results indicated that 15-0 degrees could gently decrease power by 1.50% power at 25-650 rpm compared with the 0-0 degrees condition.




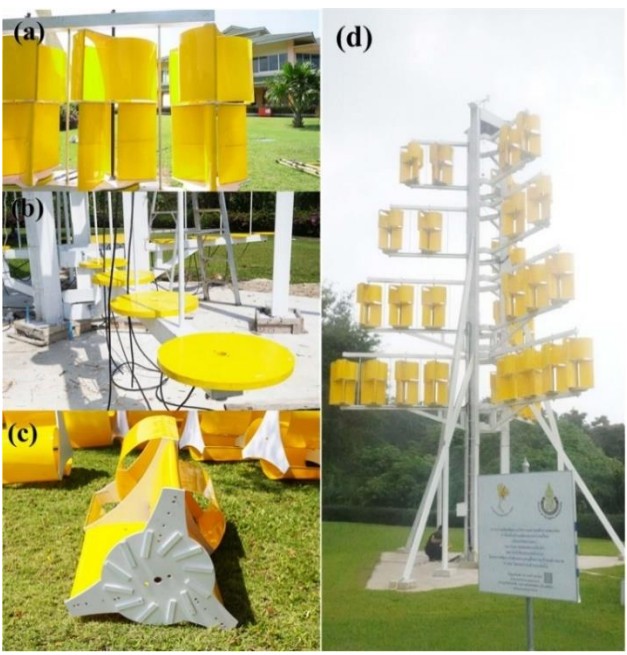

**Figure 6:** The 300 W (per turbine) vertical wind turbine installed at Rajjaprabha dam, Thailand (a) the blade of the vertical wind turbine (b) the stator of the PMSG generator (c) the rotor of the PMSG generator used in the skewing angle magnet-coil at 15-0 degree (d). 10kW wind tree combined with a 32 wind turbine (Suppachai et al., 2019 ).


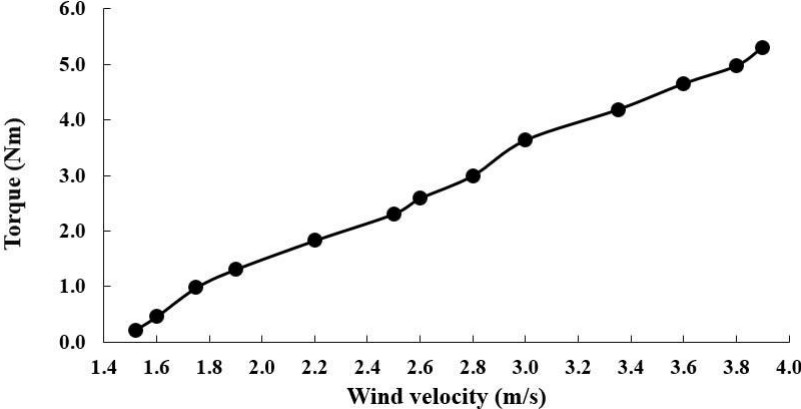

**Figure 7:** The relationship between the torque and wind speed of of the Savonius and H-Darrieus stacked blade, which is represented in **Figure 6 (a)** (Suppachai et al., 2019 ).


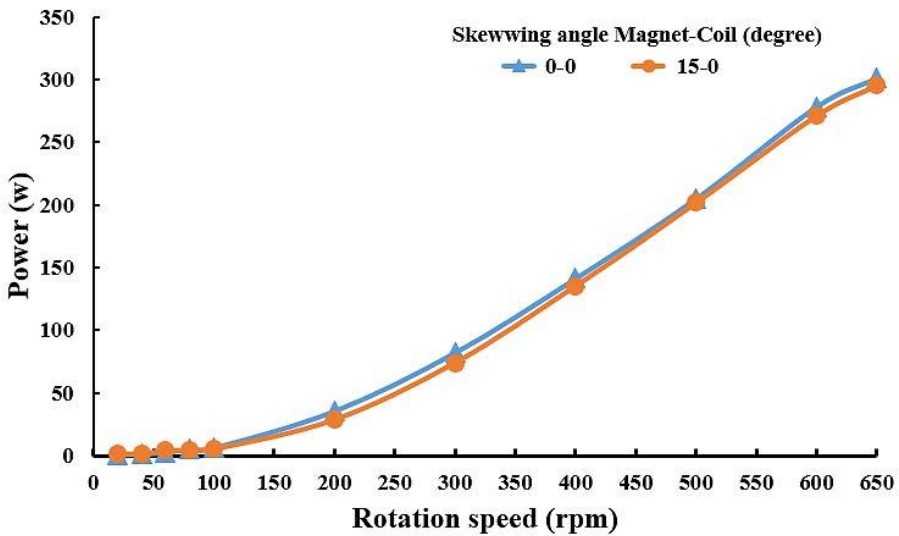

**Figure 8:** The comparison of the 0-0 and 15-0 degree skewing angle magnet-coil conditions based on the relationship of power and
rotation speed.

**Conclusion**

      The low speed vertical wind turbine requires a novel blade and electrical generator design for a cut-in at 2 m/s. The
Savonius and H-Darrieus stacked blade can be operated at a wind speed of 1.90 m/s and torque of 1.31 N-m under no-load.
Despite the PMSG electrical generator being used, it was able to generate a high starting torque that is not typically found in
a low speed wind turbine. The connection of the skewing angle magnet-coil at 15-0 degree with the blade was shown to cut-
in at 2.1 m/s in a fully operation vertical wind turbine. However, the skewing angle magnet-coil at 15-0 degree is also reduced
average power by 1.50% at 25-650 rpm. Therefore, the low speed vertical wind turbine requires a newly designed blade for
high and low starting torque PMSG generators.

**Acknowledgements**

The work is financially supported by Electricity Generating Authority of Thailand: EGAT (61-F405000-11-
IO.SS03F3008347), Energy policy and planning office (EPPO) 2018; Ministry of Energy (Thailand), Wind Turbine and
Energy Storage Systems Centre (WEESYC), Centre of Excellence in Materials Engineering (CEME), Prince of Songkla
University.





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
