# Peer review of "Skewing angle magnet and coil reduced starting torque in a permanent magnet synchronous generator for a small vertical axis wind turbine"

_Wind Energy Science, 2020_

## Referee Comment (RC1) · 12 Jan 2021

The generator discussed in the paper is an axial gap, direct-drive machine, and is targeted at small vertical axis wind turbines. This becomes clear from the reading. The results presented in the paper are principally empirical and there is little derivation or analytical development.

There are a couple of fo dubious statements that could be removed without injuring the technical aspects of the paper and which would add some credibility to the authors. Sentence at line 46, beginning with "Presently," is one example. The sentence at line 32 beginning with ""In utilizing future....," is another example. Both statements will

generate a lot of debate and neither topic should be considered settled. Likely, these topics will be settled in a direction opposite what the authors claim.

---

## Referee Comment (RC2) · A. Abrahamsen (Referee) · 16 Feb 2021

I have read the paper on the test of skew angles of the magnets and coils of a permanent magnet generator for a small scale vertical axis wind turbine (VAWT).

This paper is interesting, since it is addressing a power range below 10 kW and with the intension to be installed in placed where utility scale turbines, might not be installed in for instance remote islands in countries with a weak power grid structure. The authors argues that vertical axis wind turbiones will become relevant for large offshore wind farms, but maybe it will be better to argues for the island installation case.

The vertical axis wind turbine being addresses is a combination of a Darrieus (H-type) rotor connected on the same shaft as a Savonius type rotor in order to have a positive starting torque of the turbine. The authors are then addressing the problem of constructing a permanent magnet generator with a sufficient small starting torque to allow the turbine to spin up at a cut-in wind speed of 2 m/s.

The paper is first explaining how a test rig consisting of a motor, torque meter, a test permanent magnet generator, a variable loads and measurements equipment was constructed. Then the construction of the permanent magnet generator is introduced and the test results are presented together with a discussion.

My first overall recommendations are

1) To split the results and the discussion into two different sections in the paper. I think the discussion will be more clear is all results are presented first and then they are discussed in a separate section. 2) The authors have to explain the experimental procedure more clearly in terms of what the rotation speed is during the presented test results. This is especially the case for figure figure 3 and figure 4. 3) For vertical axis wind turbines there is often a linked relation between the turbine shaft power and both the incoming wind speed as well as the turbine rotation speed. Thus there is often a wind speed vs. rotation speed curve giving the optimal tip speed ratio for a specific turbine rotor. This relation is not clear from figure 7 and 8. Thus it seems that the rotations speed reported in figure 8 are quite high for the turbine rotor. It was not possible to find figure 7 in the reference stated as Suppachai et. al. 2019 and it was therefore hard to check. Please clarify if Suppachai et. al. 2019 is correct. It will be appropriate to provide a description of what the turbine is going to do when it is started at a wind speed of $v = 2$ m/s. Do you plan to spin up the rotation speed without the electrical load connected and then increase the electrical load at rated rotation speed? Or do you plan to connect a constant electrical load and spin up the turbine with the load connected? It will be good if this is discussed in relation to the start up torque measurements presented in figure 3 and 4.

A pdf version of the paper is attached with comments and recommendations to improvements and clarifications marked with yellow boxes. I hope this will be a help to implement the overall recommendations.

Best Regards

Asger B. Abrahamsen Senior Researcher DTU Wind Energy, Technical University of Denmark

Please also note the supplement to this comment:
https://wes.copernicus.org/preprints/wes-2020-101/wes-2020-101-RC2-supplement.pdf

**Supplement:**

[revised manuscript text omitted]

---

## Author Comment (AC1) · 28 Feb 2021

william erdman (Referee)

bill.erdman52@gmail.com

 Comments There are a couple of dubious statements that could be removed without injuring the technical aspects of the paper and which would add some credibility to the authors. - Sentence at line 46, beginning with "Presently," is one example. - The sentence at line 32 beginning with ""In utilizing future....," is another example. Both statements

will generate a lot of debate and neither topic should be considered settled. Likely, these topics will be settled in a direction opposite what the authors claim.

........................................................................................................................................

Edited by Montri Luengchavanon and the team

This article have deleted the sentence at line 32 as below:

- In utilizing future offshore wind energy, the vertical axis wind turbine (VAWT) is expected to perform better than the horizontal axis wind turbine (HAWT). This is because the VAWT has a lower center of gravity and simple mechanical structure, which is easy to fabricate, and requires low maintenance. Moreover, the VAWT has a high potential to be scaled to a number of different sizes (Paquette and Barone, 2012).

For protection the generated a lot of debate between both statements.

Please also note the supplement to this comment:
https://wes.copernicus.org/preprints/wes-2020-101/wes-2020-101-AC1-supplement.pdf

---

## Author Comment (AC2) · 28 Feb 2021

A. Abrahamsen (Referee) asab@dtu.dk Received and published: 16 February 2021 Comments 1) To split the results and the discussion into two different sections in the paper. I think the discussion will be more clear is all results are presented first and then they are discussed in a separate section. 2) The authors have to explain the experimental procedure more clearly in terms of what the rotation speed is during the presented test results. This is especially the case for figure 3 and figure 4. 3) For vertical axis wind turbines there is often a linked relation between the turbine shaft power and both the incoming wind speed as well as the turbine rotation speed. Thus

there is often a wind speed vs. rotation speed curve giving the optimal tip speed ratio for a specific turbine rotor.

3.1 This relation is not clear from figure 7 and 8. Thus it seems that the rotations speed reported in figure 8 are quite high for the turbine rotor. It was not possible to find figure 7 in the reference stated as Suppachai et. al. 2019 and it was therefore hard to check. Please clarify if Suppachai et. al. 2019 is correct.

3.2 It will be appropriate to provide a description of what the turbine is going to do when it is started at a wind speed of v = 2 m/s.

3.3 Do you plan to spin up the rotation speed without the electrical load connected and then increase the electrical load at rated rotation speed?

3.4 Or do you plan to connect a constant electrical load and spin up the turbine with the load connected?

| 3.5    | lt | will  | be | good    | if   | this | is    | discu | ssed | in    | relat | ion | to  | the |
|--------|----|-------|----|---------|------|------|-------|-------|------|-------|-------|-----|-----|-----|
| start- | up | torqu | le | measure | emer | nts  | prese | nted  | in   | figur | е     | 3   | and | 4.  |

Edited by Montri Luengchavanon and the team

- This article have already edited that split the results and discussion - All the rotational speed of 650 rpm, the maximum power generated around 300 W, but it is depend on the skewing magnet and coils technique. - This article explained more the relationship between Figure 4 (b) [Orinal Figure 3(b)] and Figure 5 [Original Figure 4] by "Fig 5 shows comparison values between the starting torque that collected peaks value from Figure 4 (b) and power when changing the skewing angle magnet-coils in the PMSG generator."

3.1 Due to the results from Suppachai et. al. 2019 used a spring torque and the wind tunnel is limited. So, this article was installed standard torque meter with high speed wind tunnel (0-10 m/s) and Savonius and H-Darrieus wind turbine, this article

WESD
was added Figure.3. For explained the relationship between figure 8 and 9 (original Figure 7 and 8) that Figure 8 shows the related between rotation speed, torque and wind speed. Figure 9 is related to Figure 8 by rotation speed which used skewing angle magnet-coil at 15-0 degree, it can be really reduced starting torque and cut-in at 2.1 m/s.

3.2 The cut-in of this vertical wind turbine cannot start turning at 2.0 m/s wind speed due to the systems combined force of Tstarting, friction and weight of accessory. Hence, the real operation of this vertical axis wind turbine system used (Savonius and H-Daarieus) and skewing magnet-coil at 15-0 degree that can be cut-in at 2.1 m/s as shown in Figure 8 [Original Figure 7].

(3.3 and 3.4) Yes, this experiment was firstly spin up the rotation speed without load and then increased the electrical load at many rated rotation speed.

3.5 Figure 4(a) [original Figure 3 (a)] represents the starting torque are without electrical load. Figure 4(b) [original Figure 3 (b)] represented the starting torque are connected the electrical load. Figure 5 [Original Figure 4] is related Figure 4(b) [original Figure 3 (b)] which Figure 5 was compared the maximum peaks of starting-up torque at all the skewing magnet – coils that looking for high power and low start-up torque.

Please also note the supplement to this comment: https://wes.copernicus.org/preprints/wes-2020-101/wes-2020-101-AC2supplement.pdf